# Stein Variational Gradient Descent: A General Purpose Bayesian Inference Algorithm

**Qiang Liu**     **Dilin Wang**
Department of Computer Science
Dartmouth College
Hanover, NH 03755
{qiang.liu, dilin.wang.gr}@dartmouth.edu

## Abstract

We propose a general purpose variational inference algorithm that forms a natural counterpart of gradient descent for optimization. Our method iteratively transports a set of particles to match the target distribution, by applying a form of functional gradient descent that minimizes the KL divergence. Empirical studies are performed on various real world models and datasets, on which our method is competitive with existing state-of-the-art methods. The derivation of our method is based on a new theoretical result that connects the derivative of KL divergence under smooth transforms with Stein's identity and a recently proposed kernelized Stein discrepancy, which is of independent interest.

## 1 Introduction

Bayesian inference provides a powerful tool for modeling complex data and reasoning under uncertainty, but casts a long standing challenge on computing intractable posterior distributions. Markov chain Monte Carlo (MCMC) has been widely used to draw approximate posterior samples, but is often slow and has difficulty accessing the convergence. Variational inference instead frames the Bayesian inference problem into a deterministic optimization that approximates the target distribution with a simpler distribution by minimizing their KL divergence. This makes variational methods efficiently solvable by using off-the-shelf optimization techniques, and easily applicable to large datasets (i.e., "big data") using the stochastic gradient descent trick [e.g., 1]. In contrast, it is much more challenging to scale up MCMC to big data settings [see e.g., 2, 3].

Meanwhile, both the accuracy and computational cost of variational inference critically depend on the set of distributions in which the approximation is defined. Simple approximation sets, such as these used in the traditional mean field methods, are too restrictive to resemble the true posterior distributions, while more advanced choices cast more difficulties on the subsequent optimization tasks. For this reason, efficient variational methods often need to be derived on a model-by-model basis, causing is a major barrier for developing general purpose, user-friendly variational tools applicable for different kinds of models, and accessible to non-ML experts in application domains.

This case is in contrast with the maximum *a posteriori* (MAP) optimization tasks for finding the posterior mode (sometimes known as the *poor man's Bayesian estimator*, in contrast with the *full Bayesian inference* for approximating the full posterior distribution), for which variants of (stochastic) gradient descent serve as a simple, generic, yet extremely powerful toolbox. There has been a recent growth of interest in creating user-friendly variational inference tools [e.g., 4–7], but more efforts are still needed to develop more efficient general purpose algorithms.

In this work, we propose a new general purpose variational inference algorithm which can be treated as a natural counterpart of gradient descent for full Bayesian inference (see Algorithm 1). Our

algorithm uses a set of particles for approximation, on which a form of (functional) gradient descent is performed to minimize the KL divergence and drive the particles to fit the true posterior distribution. Our algorithm has a simple form, and can be applied whenever gradient descent can be applied. In fact, it reduces to gradient descent for MAP when using only a single particle, while automatically turns into a full Bayesian sampling approach with more particles.

Underlying our algorithm is a new theoretical result that connects the derivative of KL divergence w.r.t. smooth variable transforms and a recently introduced kernelized Stein discrepancy [8–10], which allows us to derive a closed form solution for the optimal smooth perturbation direction that gives the steepest descent on the KL divergence within the unit ball of a reproducing kernel Hilbert space (RKHS). This new result is of independent interest, and can find wide application in machine learning and statistics beyond variational inference.

## 2   Background

**Preliminary**   Let $x$ be a continuous random variable or parameter of interest taking values in $\mathcal{X} \subset \mathbb{R}^d$, and $\{D_k\}$ is a set of i.i.d. observation. With prior $p_0(x)$, Bayesian inference of $x$ involves reasoning with the posterior distribution $p(x) := \bar{p}(x)/Z$ with $\bar{p}(x) := p_0(x) \prod_{k=1}^N p(D_k|x)$, where $Z = \int \bar{p}(x)dx$ is the troublesome normalization constant. We have dropped the conditioning on data $\{D_k\}$ in $p(x)$ for convenience.

Let $k(x, x') \colon \mathcal{X} \times \mathcal{X} \to \mathbb{R}$ be a positive definite kernel. The reproducing kernel Hilbert space (RKHS) $\mathcal{H}$ of $k(x, x')$ is the closure of linear span $\{f \colon f(x) = \sum_{i=1}^m a_i k(x, x_i), \;\; a_i \in \mathbb{R}, \;\; m \in \mathbb{N}, \; x_i \in \mathcal{X}\}$, equipped with inner products $\langle f, \; g \rangle_{\mathcal{H}} = \sum_{ij} a_i b_j k(x_i, x_j)$ for $g(x) = \sum_i b_i k(x, x_i)$. Denote by $\mathcal{H}^d$ the space of vector functions $\boldsymbol{f} = [f_1, \ldots, f_d]^\top$ with $f_i \in \mathcal{H}$, equipped with inner product $\langle \boldsymbol{f}, \boldsymbol{g} \rangle_{\mathcal{H}^d} = \sum_{i=1}^d \langle f_i, g_i \rangle_{\mathcal{H}}$. We assume all the vectors are column vectors. Let $\nabla_x \boldsymbol{f} = [\nabla_x f_1, \ldots, \nabla_x f_d]$.

**Stein's Identity and Kernelized Stein Discrepancy**   Stein's identity plays a fundamental role in our framework. Let $p(x)$ be a continuously differentiable (also called smooth) density supported on $\mathcal{X} \subseteq \mathbb{R}^d$, and $\boldsymbol{\phi}(x) = [\phi_1(x), \cdots, \phi_d(x)]^\top$ a smooth vector function. Stein's identity states that for sufficiently regular $\boldsymbol{\phi}$, we have

$$\mathbb{E}_{x \sim p}[\mathcal{A}_p \boldsymbol{\phi}(x)] = 0, \qquad \text{where} \qquad \mathcal{A}_p \boldsymbol{\phi}(x) = \nabla_x \log p(x) \boldsymbol{\phi}(x)^\top + \nabla_x \boldsymbol{\phi}(x), \qquad (1)$$

where $\mathcal{A}_p$ is called the Stein operator, which acts on function $\boldsymbol{\phi}$ and yields a zero mean function $\mathcal{A}_p \boldsymbol{\phi}(x)$ under $x \sim p$. This identity can be easily checked using integration by parts by assuming mild zero boundary conditions on $\boldsymbol{\phi}$: either $p(x)\boldsymbol{\phi}(x) = 0$, $\forall x \in \partial \mathcal{X}$ when $\mathcal{X}$ is compact, or $\lim_{r \to \infty} \oint_{B_r} p(x)\boldsymbol{\phi}(x)^\top \boldsymbol{n}(x)dS = 0$ when $\mathcal{X} = \mathbb{R}^d$, where $\oint_{B_r}$ is the surface integral on the sphere $B_r$ of radius $r$ centered at the origin and $\boldsymbol{n}(x)$ is the unit normal to $B_r$. We call that $\boldsymbol{\phi}$ is in the Stein class of $p$ if Stein's identity (1) holds.

Now let $q(x)$ be a different smooth density also supported in $\mathcal{X}$, and consider the expectation of $\mathcal{A}_p \boldsymbol{\phi}(x)$ under $x \sim q$, then $\mathbb{E}_{x \sim q}[\mathcal{A}_p \boldsymbol{\phi}(x)]$ would no longer equal zero for general $\boldsymbol{\phi}$. Instead, the magnitude of $\mathbb{E}_{x \sim q}[\mathcal{A}_p \boldsymbol{\phi}(x)]$ relates to how different $p$ and $q$ are, and can be leveraged to define a discrepancy measure, known as *Stein discrepancy*, by considering the "maximum violation of Stein's identity" for $\boldsymbol{\phi}$ in some proper function set $\mathcal{F}$:

$$\mathbb{D}(q, \; p) = \max_{\boldsymbol{\phi} \in \mathcal{F}} \{ \mathbb{E}_{x \sim q}[\text{trace}(\mathcal{A}_p \boldsymbol{\phi}(x))] \},$$

Here the choice of this function set $\mathcal{F}$ is critical, and decides the discriminative power and computational tractability of Stein discrepancy. Traditionally, $\mathcal{F}$ is taken to be sets of functions with bounded Lipschitz norms, which unfortunately casts a challenging functional optimization problem that is computationally intractable or requires special considerations (see Gorham and Mackey [11] and reference therein).

Kernelized Stein discrepancy (KSD) bypasses this difficulty by maximizing $\boldsymbol{\phi}$ in the unit ball of a reproducing kernel Hilbert space (RKHS) for which the optimization has a closed form solution. KSD is defined as

$$\mathbb{D}(q, \; p) = \max_{\boldsymbol{\phi} \in \mathcal{H}^d} \{ \mathbb{E}_{x \sim q}[\text{trace}(\mathcal{A}_p \boldsymbol{\phi}(x))], \quad s.t. \quad ||\boldsymbol{\phi}||_{\mathcal{H}^d} \leq 1 \}, \qquad (2)$$

where we assume the kernel $k(x, x')$ of RKHS $\mathcal{H}$ is in the Stein class of $p$ as a function of $x$ for any fixed $x' \in \mathcal{X}$. The optimal solution of (2) has been shown to be $\phi(x) = \phi_{q,p}^*(x)/||\phi_{q,p}^*||_{\mathcal{H}^d}$ [8–10], where

$$\phi_{q,p}^*(\cdot) = \mathbb{E}_{x \sim q}[\mathcal{A}_p k(x, \cdot)], \qquad \text{for which we have} \qquad \mathbb{D}(q, \ p) = ||\phi_{q,p}^*||_{\mathcal{H}^d}. \qquad (3)$$

One can further show that $\mathbb{D}(q, \ p)$ equals zero (and equivalently $\phi_{q,p}^*(x) \equiv 0$) if and only if $p = q$ once $k(x, x')$ is strictly positive definite in a proper sense [See 8, 10], which is satisfied by commonly used kernels such as the RBF kernel $k(x, x') = \exp(-\frac{1}{h}||x - x'||_2^2)$. Note that the RBF kernel is also in the Stein class of smooth densities supported in $\mathcal{X} = \mathbb{R}^d$ because of its decaying property.

Both Stein operator and KSD depend on $p$ only through the score function $\nabla_x \log p(x)$, which can be calculated without knowing the normalization constant of $p$, because we have $\nabla_x \log p(x) = \nabla_x \log \bar{p}(x)$ when $p(x) = \bar{p}(x)/Z$. This property makes Stein's identity a powerful tool for handling unnormalized distributions that appear widely in machine learning and statistics.

## 3 Variational Inference Using Smooth Transforms

Variational inference approximates the target distribution $p(x)$ using a simpler distribution $q^*(x)$ found in a predefined set $\mathcal{Q} = \{q(x)\}$ of distributions by minimizing the KL divergence, that is,

$$q^* = \underset{q \in \mathcal{Q}}{\arg\min} \left\{ \mathrm{KL}(q \ || \ p) \equiv \mathbb{E}_q[\log q(x)] - \mathbb{E}_q[\log \bar{p}(x)] + \log Z \right\}, \qquad (4)$$

where we do not need to calculate the constant $\log Z$ for solving the optimization. The choice of set $\mathcal{Q}$ is critical and defines different types of variational inference methods. The best set $\mathcal{Q}$ should strike a balance between i) *accuracy*, broad enough to closely approximate a large class of target distributions, ii) *tractability*, consisting of simple distributions that are easy for inference, and iii) *solvability* so that the subsequent KL minimization problem can be efficiently solved.

In this work, we focus on the sets $\mathcal{Q}$ consisting of distributions obtained by smooth transforms from a tractable reference distribution, that is, we take $\mathcal{Q}$ to be the set of distributions of random variables of form $z = \boldsymbol{T}(x)$ where $\boldsymbol{T} \colon \mathcal{X} \to \mathcal{X}$ is a smooth one-to-one transform, and $x$ is drawn from a tractable reference distribution $q_0(x)$. By the change of variables formula, the density of $z$ is

$$q_{[\boldsymbol{T}]}(z) = q(\boldsymbol{T}^{-1}(z)) \cdot |\det(\nabla_z \boldsymbol{T}^{-1}(z))|,$$

where $\boldsymbol{T}^{-1}$ denotes the inverse map of $\boldsymbol{T}$ and $\nabla_z \boldsymbol{T}^{-1}$ the Jacobian matrix of $\boldsymbol{T}^{-1}$. Such distributions are computationally tractable, in the sense that the expectation under $q_{[\boldsymbol{T}]}$ can be easily evaluated by averaging $\{z_i\}$ when $z_i = \boldsymbol{T}(x_i)$ and $x_i \sim q_0$. Such $\mathcal{Q}$ can also in principle closely approximate almost arbitrary distributions: it can be shown that there always exists a measurable transform $\boldsymbol{T}$ between any two distributions without atoms (i.e. no single point carries a positive mass); in addition, for Lipschitz continuous densities $p$ and $q$, there always exist transforms between them that are least as smooth as both $p$ and $q$. We refer the readers to Villani [12] for in-depth discussion on this topic.

In practice, however, we need to restrict the set of transforms $\boldsymbol{T}$ properly to make the corresponding variational optimization in (4) practically solvable. One approach is to consider $\boldsymbol{T}$ with certain parametric form and optimize the corresponding parameters [e.g., 13, 14]. However, this introduces a difficult problem on selecting the proper parametric family to balance the accuracy, tractability and solvability, especially considering that $\boldsymbol{T}$ has to be an one-to-one map and has to have an efficiently computable Jacobian matrix.

Instead, we propose a new algorithm that iteratively constructs incremental transforms that effectively perform steepest descent on $\boldsymbol{T}$ in RKHS. Our algorithm does not require to explicitly specify parametric forms, nor to calculate the Jacobian matrix, and has a particularly simple form that mimics the typical gradient descent algorithm, making it easily implementable even for non-experts in variational inference.

### 3.1 Stein Operator as the Derivative of KL Divergence

To explain how we minimize the KL divergence in (4), we consider an incremental transform formed by a small perturbation of the identity map: $\boldsymbol{T}(x) = x + \epsilon \phi(x)$, where $\phi(x)$ is a smooth function

that characterizes the perturbation direction and the scalar $\epsilon$ represents the perturbation magnitude. When $|\epsilon|$ is sufficiently small, the Jacobian of $\boldsymbol{T}$ is full rank (close to the identity matrix), and hence $\boldsymbol{T}$ is guaranteed to be an one-to-one map by the inverse function theorem.

The following result, which forms the foundation of our method, draws an insightful connection between Stein operator and the derivative of KL divergence w.r.t. the perturbation magnitude $\epsilon$.

**Theorem 3.1.** *Let $\boldsymbol{T}(x) = x + \epsilon\boldsymbol{\phi}(x)$ and $q_{[\boldsymbol{T}]}(z)$ the density of $z = \boldsymbol{T}(x)$ when $x \sim q(x)$, we have*

$$\nabla_\epsilon \mathrm{KL}(q_{[\boldsymbol{T}]} \,\|\, p)\,\big|_{\epsilon=0} = -\mathbb{E}_{x \sim q}[\mathrm{trace}(\mathcal{A}_p\boldsymbol{\phi}(x))], \tag{5}$$

*where $\mathcal{A}_p\boldsymbol{\phi}(x) = \nabla_x \log p(x)\boldsymbol{\phi}(x)^\top + \nabla_x\boldsymbol{\phi}(x)$ is the Stein operator.*

Relating this to the definition of KSD in (2), we can identify the $\boldsymbol{\phi}^*_{q,p}$ in (3) as the optimal perturbation direction that gives the steepest descent on the KL divergence in zero-centered balls of $\mathcal{H}^d$.

**Lemma 3.2.** *Assume the conditions in Theorem 3.1. Consider all the perturbation directions $\boldsymbol{\phi}$ in the ball $\mathcal{B} = \{\boldsymbol{\phi} \in \mathcal{H}^d : \|\boldsymbol{\phi}\|_{\mathcal{H}^d} \le \mathbb{D}(q,\,p)\}$ of vector-valued RKHS $\mathcal{H}^d$, the direction of steepest descent that maximizes the negative gradient in (5) is the $\boldsymbol{\phi}^*_{q,p}$ in (3), i.e.,*

$$\boldsymbol{\phi}^*_{q,p}(\cdot) = \mathbb{E}_{x \sim q}[\nabla_x \log p(x)k(x,\cdot) + \nabla_x k(x,\cdot)], \tag{6}$$

*for which (5) equals the square of KSD, that is, $\nabla_\epsilon \mathrm{KL}(q_{[\boldsymbol{T}]} \,\|\, p)\,\big|_{\epsilon=0} = -\mathbb{D}^2(q,\,p)$.*

The result in Lemma (3.2) suggests an iterative procedure that transforms an initial reference distribution $q_0$ to the target distribution $p$: we start with applying transform $\boldsymbol{T}^*_0(x) = x + \epsilon_0 \cdot \boldsymbol{\phi}^*_{q_0,p}(x)$ on $q_0$ which decreases the KL divergence by an amount of $\epsilon_0 \cdot \mathbb{D}^2(q_0,\,p)$, where $\epsilon_0$ is a small step size; this would give a new distribution $q_1(x) = q_{0[\boldsymbol{T}_0]}(x)$, on which a further transform $\boldsymbol{T}^*_1(x) = x + \epsilon_1 \cdot \boldsymbol{\phi}^*_{q_1,p}(x)$ can further decrease the KL divergence by $\epsilon_1 \cdot \mathbb{D}^2(q_1,\,p)$. Repeating this process one constructs a path of distributions $\{q_\ell\}_{\ell=1}^n$ between $q_0$ and $p$ via

$$q_{\ell+1} = q_{\ell[\boldsymbol{T}^*_\ell]}, \qquad \text{where} \qquad \boldsymbol{T}^*_\ell(x) = x + \epsilon_\ell \cdot \boldsymbol{\phi}^*_{q_\ell,p}(x). \tag{7}$$

This would eventually converge to the target $p$ with sufficiently small step-size $\{\epsilon_\ell\}$, under which $\boldsymbol{\phi}^*_{p,q_\infty}(x) \equiv 0$ and $\boldsymbol{T}^*_\infty$ reduces to the identity map. Recall that $q_\infty = p$ if and only if $\boldsymbol{\phi}^*_{p,q_\infty}(x) \equiv 0$.

**Functional Gradient**   To gain further intuition on this process, we now reinterpret (6) as a functional gradient in RKHS. For any functional $F[\boldsymbol{f}]$ of $\boldsymbol{f} \in \mathcal{H}^d$, its (functional) gradient $\nabla_{\boldsymbol{f}}F[\boldsymbol{f}]$ is a function in $\mathcal{H}^d$ such that $F[\boldsymbol{f} + \epsilon\boldsymbol{g}(x)] = F[\boldsymbol{f}] + \epsilon \langle \nabla_{\boldsymbol{f}}F[\boldsymbol{f}],\, \boldsymbol{g} \rangle_{\mathcal{H}^d} + O(\epsilon^2)$ for any $\boldsymbol{g} \in \mathcal{H}^d$ and $\epsilon \in \mathbb{R}$.

**Theorem 3.3.** *Let $\boldsymbol{T}(x) = x + \boldsymbol{f}(x)$, where $\boldsymbol{f} \in \mathcal{H}^d$, and $q_{[\boldsymbol{T}]}$ the density of $z = \boldsymbol{T}(x)$ when $x \sim q$,*

$$\nabla_{\boldsymbol{f}}\mathrm{KL}(q_{[\boldsymbol{T}]} \,\|\, p)\,\big|_{\boldsymbol{f}=0} = -\boldsymbol{\phi}^*_{q,p}(x),$$

*whose RKHS norm is $\|\boldsymbol{\phi}^*_{q,p}\|_{\mathcal{H}^d} = \mathbb{D}(q,p)$.*

This suggests that $\boldsymbol{T}^*(x) = x + \epsilon \cdot \boldsymbol{\phi}^*_{q,p}(x)$ is equivalent to a step of functional gradient descent in RKHS. However, what is critical in the iterative procedure (7) is that we also iteratively apply the variable transform so that every time we would only need to evaluate the functional gradient descent at zero perturbation $\boldsymbol{f} = 0$ on the identity map $\boldsymbol{T}(x) = x$. This brings a critical advantage since the gradient at $\boldsymbol{f} \ne 0$ is more complex and would require to calculate the inverse Jacobian matrix $[\nabla_x\boldsymbol{T}(x)]^{-1}$ that casts computational or implementation hurdles.

## 3.2   Stein Variational Gradient Descent

To implement the iterative procedure (7) in practice, one would need to approximate the expectation for calculating $\boldsymbol{\phi}^*_{q,p}(x)$ in (6). To do this, we can first draw a set of particles $\{x^0_i\}_{i=1}^n$ from the initial distribution $q_0$, and then iteratively update the particles with an empirical version of the transform in (7) in which the expectation under $q_\ell$ in $\boldsymbol{\phi}^*_{q_\ell,p}$ is approximated by the empirical mean of particles $\{x^\ell_i\}_{i=1}^n$ at the $\ell$-th iteration. This procedure is summarized in Algorithm 1, which allows us to (deterministically) transport a set of points to match our target distribution $p(x)$, effectively providing

---

**Algorithm 1** Bayesian Inference via Variational Gradient Descent

---

**Input:** A target distribution with density function $p(x)$ and a set of initial particles $\{x_i^0\}_{i=1}^n$.
**Output:** A set of particles $\{x_i\}_{i=1}^n$ that approximates the target distribution $p(x)$.
**for** iteration $\ell$ **do**

$$x_i^{\ell+1} \leftarrow x_i^\ell + \epsilon_\ell \hat{\phi}^*(x_i^\ell) \quad \text{where} \quad \hat{\phi}^*(x) = \frac{1}{n} \sum_{j=1}^n \left[ k(x_j^\ell, x) \nabla_{x_j^\ell} \log p(x_j^\ell) + \nabla_{x_j^\ell} k(x_j^\ell, x) \right], \quad (8)$$

where $\epsilon_\ell$ is the step size at the $\ell$-th iteration.
**end for**

---

a sampling method for $p(x)$. We can see that the implementation of this procedure does not depend on the initial distribution $q_0$ at all, and in practice we can start with a set of arbitrary points $\{x_i\}_{i=1}^n$, possibly generated by a complex (randomly or deterministic) black-box procedure.

We can expect that $\{x_i^\ell\}_{i=1}^n$ forms increasingly better approximation for $q_\ell$ as $n$ increases. To see this, denote by $\Phi$ the nonlinear map that takes the measure of $q_\ell$ and outputs that of $q_{\ell+1}$ in (7), that is, $q_{\ell+1} = \Phi_\ell(q_\ell)$, where $q_\ell$ enters the map through both $q_{\ell[T_\ell^*]}$ and $\phi_{q_\ell,p}^*$. Then, the updates in Algorithm 1 can be seen as applying the same map $\Phi$ on the empirical measure $\hat{q}_\ell$ of particles $\{x_i^\ell\}$ to get the empirical measure $\hat{q}_{\ell+1}$ of particles $\{x_i^{\ell+1}\}$ at the next iteration, that is, $\hat{q}_{\ell+1} = \Phi_\ell(\hat{q}_\ell)$. Since $\hat{q}_0$ converges to $q_0$ as $n$ increases, $\hat{q}_\ell$ should also converge to $q_\ell$ when the map $\Phi$ is "continuous" in a proper sense. Rigorous theoretical results on such convergence have been established in the mean field theory of interacting particle systems [e.g., 15], which in general guarantee that $\sum_{i=1}^n h(x_i^\ell)/n - \mathbb{E}_{q_\ell}[h(x)] = \mathcal{O}(1/\sqrt{n})$ for bounded testing functions $h$. In addition, the distribution of each particle $x_{i_0}^\ell$, for any fixed $i_0$, also tends to $q_\ell$, and is independent with any other finite subset of particles as $n \to \infty$, a phenomenon called *propagation of chaos* [16]. We leave concrete theoretical analysis for future work.

Algorithm 1 mimics a gradient dynamics at the particle level, where the two terms in $\hat{\phi}^*(x)$ in (8) play different roles: the first term drives the particles towards the high probability areas of $p(x)$ by following a *smoothed* gradient direction, which is the weighted sum of the gradients of all the points weighted by the kernel function. The second term acts as a *repulsive force* that prevents all the points to collapse together into local modes of $p(x)$; to see this, consider the RBF kernel $k(x, x') = \exp(-\frac{1}{h}||x - x'||^2)$, the second term reduces to $\sum_j \frac{2}{h}(x - x_j)k(x_j, x)$, which drives $x$ away from its neighboring points $x_j$ that have large $k(x_j, x)$. If we let bandwidth $h \to 0$, the repulsive term vanishes, and update (8) reduces to a set of independent chains of typical gradient ascent for maximizing $\log p(x)$ (i.e., MAP) and all the particles would collapse into the local modes.

Another interesting case is when we use only a single particle ($n = 1$), in which case Algorithm 1 reduces to a single chain of typical gradient ascent for MAP for any kernel that satisfies $\nabla_x k(x, x) = 0$ (for which RBF holds). This suggests that our algorithm can generalize well for supervised learning tasks even with a very small number $n$ of particles, since gradient ascent for MAP ($n = 1$) has been shown to be very successful in practice. This property distinguishes our particle method with the typical Monte Carlo methods that requires to average over many points. The key difference here is that we use a deterministic repulsive force, other than Monte Carlo randomness, to get diverse points for distributional approximation.

**Complexity and Efficient Implementation** The major computation bottleneck in (8) lies on calculating the gradient $\nabla_x \log p(x)$ for all the points $\{x_i\}_{i=1}^n$; this is especially the case in big data settings when $p(x) \propto p_0(x) \prod_{k=1}^N p(D_k|x)$ with a very large $N$. We can conveniently address this problem by approximating $\nabla_x \log p(x)$ with subsampled mini-batches $\Omega \subset \{1, \ldots, N\}$ of the data

$$\nabla_x \log p(x) \approx \log p_0(x) + \frac{N}{|\Omega|} \sum_{k \in \Omega} \log p(D_k \mid x). \quad (9)$$

Additional speedup can be obtained by parallelizing the gradient evaluation of the $n$ particles.

The update (8) also requires to compute the kernel matrix $\{k(x_i, x_j)\}$ which costs $\mathcal{O}(n^2)$; in practice, this cost can be relatively small compared with the cost of gradient evaluation, since it can be sufficient to use a relatively small $n$ (e.g., several hundreds) in practice. If there is a need for very large $n$, one

can approximate the summation $\sum_{i=1}^{n}$ in (8) by subsampling the particles, or using a random feature expansion of the kernel $k(x, x')$ [17].

# 4 Related Works

Our work is mostly related to Rezende and Mohamed [13], which also considers variational inference over the set of transformed random variables, but focuses on transforms of parametric form $T(x) = f_\ell(\cdots(f_1(f_0(x))))$ where $f_i(\cdot)$ is a predefined simple parametric transform and $\ell$ a predefined length; this essentially creates a feedforward neural network with $\ell$ layers, whose invertibility requires further conditions on the parameters and needs to be established case by case. The similar idea is also discussed in Marzouk et al. [14], which also considers transforms parameterized in special ways to ensure the invertible and the computational tractability of the Jacobian matrix. Recently, Tran et al. [18] constructed a variational family that achieves universal approximation based on Gaussian process (equivalent to a single-layer, infinitely-wide neural network), which does not have a Jacobian matrix but needs to calculate the inverse of the kernel matrix of the Gaussian process. Our algorithm has a simpler form, and does not require to calculate any matrix determinant or inversion. Several other works also leverage variable transforms in variational inference, but with more limited forms; examples include affine transforms [19, 20], and recently the copula models that correspond to element-wise transforms over the individual variables [21, 22].

Our algorithm maintains and updates a set of particles, and is of similar style with the Gaussian mixture variation inference methods whose mean parameters can be treated as a set of particles. [23–26, 5]. Optimizing such mixture KL objectives often requires certain approximation, and this was done most recently in Gershman et al. [5] by approximating the entropy using Jensen's inequality and the expectation term using Taylor approximation. There is also a large set of particle-based Monte Carlo methods, including variants of sequential Monte Carlo [e.g., 27, 28], as well as a recent particle mirror descent for optimizing the variational objective function [7]; compared with these methods, our method does not have the weight degeneration problem, and is much more "particle-efficient" in that we reduce to MAP with only one single particle.

# 5 Experiments

We test our algorithm on both toy and real world examples, on which we find our method tends to outperform a variety of baseline methods. Our code is available at `https://github.com/DartML/Stein-Variational-Gradient-Descent`.

For all our experiments, we use RBF kernel $k(x, x') = \exp(-\frac{1}{h}||x - x'||_2^2)$, and take the bandwidth to be $h = \mathrm{med}^2 / \log n$, where $\mathrm{med}$ is the median of the pairwise distance between the current points $\{x_i\}_{i=1}^{n}$; this is based on the intuition that we would have $\sum_j k(x_i, x_j) \approx n \exp(-\frac{1}{h}\mathrm{med}^2) = 1$, so that for each $x_i$ the contribution from its own gradient and the influence from the other points balance with each other. Note that in this way, the bandwidth $h$ actually changes adaptively across the iterations. We use AdaGrad for step size and initialize the particles using the prior distribution unless otherwise specified.

**Toy Example on 1D Gaussian Mixture**  We set our target distribution to be $p(x) = 1/3\mathcal{N}(x; -2, 1) + 2/3\mathcal{N}(x; 2, 1)$, and initialize the particles using $q_0(x) = \mathcal{N}(x; -10, 1)$. This creates a challenging situation since the probability mass of $p(x)$ and $q_0(x)$ are far away each other (with almost zero overlap). Figure 1 shows how the distribution of the particles ($n = 1$) of our method evolve at different iterations. We see that despite the small overlap between $q_0(x)$ and $p(x)$, our method can push the particles towards the target distribution, and even recover the mode that is further away from the initial point. We found that other particle based algorithms, such as Dai et al. [7], tend to experience weight degeneracy on this toy example due to the ill choice of $q_0(x)$.

Figure 2 compares our method with Monte Carlo sampling when using the obtained particles to estimate expectation $\mathbb{E}_p(h(x))$ with different test functions $h(\cdot)$. We see that the MSE of our method tends to perform similarly or better than the exact Monte Carlo sampling. This may be because our particles are more spread out than i.i.d. samples due to the repulsive force, and hence give higher estimation accuracy. It remains an open question to formally establish the error rate of our method.

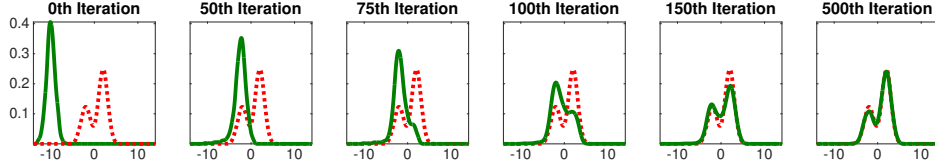

Figure 1: Toy example with 1D Gaussian mixture. The red dashed lines are the target density function and the solid green lines are the densities of the particles at different iterations of our algorithm (estimated using kernel density estimator) . Note that the initial distribution is set to have almost zero overlap with the target distribution, and our method demonstrates the ability of escaping the local mode on the left to recover the mode on the left that is further away. We use $n = 100$ particles.

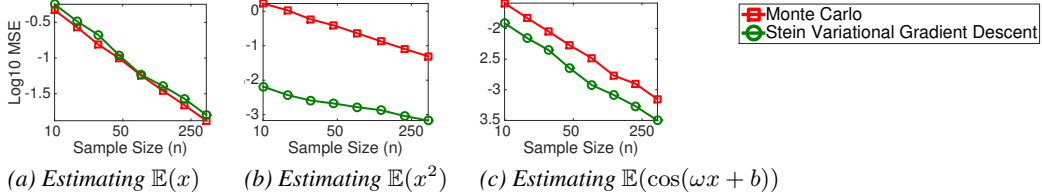

(a) Estimating $\mathbb{E}(x)$     (b) Estimating $\mathbb{E}(x^2)$     (c) Estimating $\mathbb{E}(\cos(\omega x + b))$

Figure 2: We use the same setting as Figure 1, except varying the number $n$ of particles. (a)-(c) show the mean square errors when using the obtained particles to estimate expectation $\mathbb{E}_p(h(x))$ for $h(x) = x$, $x^2$, and $\cos(\omega x + b)$; for $\cos(\omega x + b)$, we draw $\omega \sim \mathcal{N}(0, 1)$ and $b \sim \mathrm{Uniform}([0, 2\pi])$ and report the average MSE over 20 random draws of $\omega$ and $b$.

**Bayesian Logistic Regression** We consider Bayesian logistic regression for binary classification using the same setting as Gershman et al. [5], which assigns the regression weights $w$ with a Gaussian prior $p_0(w|\alpha) = \mathcal{N}(w, \alpha^{-1})$ and $p_0(\alpha) = Gamma(\alpha, 1, 0.01)$. The inference is applied on posterior $p(x|D)$ with $x = [w, \log \alpha]$. We compared our algorithm with the no-U-turn sampler (NUTS)[1] [29] and non-parametric variational inference (NPV)[2] [5] on the 8 datasets ($N > 500$) used in Gershman et al. [5], and find they tend to give very similar results on these (relatively simple) datasets; see Appendix for more details.

We further test the binary Covertype dataset[3] with 581,012 data points and 54 features. This dataset is too large, and a stochastic gradient descent is needed for speed. Because NUTS and NPV do not have mini-batch option in their code, we instead compare with the stochastic gradient Langevin dynamics (SGLD) by Welling and Teh [2], the particle mirror descent (PMD) by Dai et al. [7], and the doubly stochastic variational inference (DSVI) by Titsias and Lázaro-Gredilla [19].[4] We also compare with a parallel version of SGLD that runs $n$ parallel chains and take the last point of each chain as the result. This parallel SGLD is similar with our method and we use the same step-size of $\epsilon_\ell = a/(t+1)^{.55}$ for both as suggested by Welling and Teh [2] for fair comparison;[5] we select $a$ using a validation set within the training set. For PMD, we use a step size of $\frac{a}{N}/(100 + \sqrt{t})$, and RBF kernel $k(x, x') = \exp(-||x - x'||^2/h)$ with bandwidth $h = 0.002 \times \mathrm{med}^2$ which is based on the guidance of Dai et al. [7] which we find works most efficiently for PMD. Figure 3(a)-(b) shows the results when we initialize our method and both versions of SGLD using the prior $p_0(\alpha)p_0(w|\alpha)$; we find that PMD tends to be unstable with this initialization because it generates weights $w$ with large magnitudes, so we divided the initialized weights by 10 for PMD; as shown in Figure 3(a), this gives some advantage to PMD in the initial stage. We find our method generally performs the best, followed with the parallel SGLD, which is much better than its sequential counterpart; this comparison is of course in favor of parallel SGLD, since each iteration of it requires $n = 100$ times of likelihood evaluations compared with sequential SGLD. However, by leveraging the matrix operation in MATLAB, we find that each iteration of parallel SGLD is only 3 times more expensive than sequential SGLD.

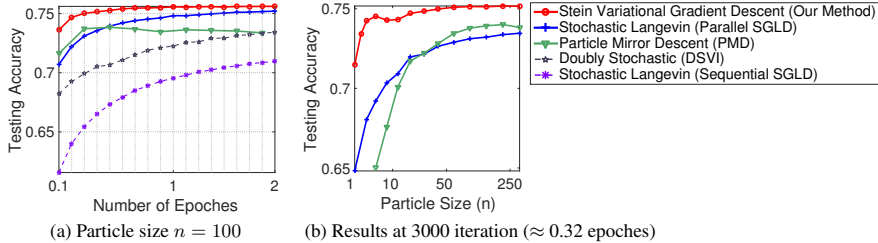

(a) Particle size $n = 100$          (b) Results at 3000 iteration ($\approx 0.32$ epochs)

Figure 3: Results on Bayesian logistic regression on Covertype dataset w.r.t. epochs and the particle size $n$. We use $n = 100$ particles for our method, parallel SGLD and PMD, and average the last 100 points for the sequential SGLD. The "particle-based" methods (solid lines) in principle require 100 times of likelihood evaluations compare with DVSI and sequential SGLD (dash lines) per iteration, but are implemented efficiently using Matlab matrix operation (e.g., each iteration of parallel SGLD is about 3 times slower than sequential SGLD). We partition the data into $80\%$ for training and $20\%$ for testing and average on 50 random trials. A mini-batch size of 50 is used for all the algorithms.

**Bayesian Neural Network** We compare our algorithm with the probabilistic back-propagation (PBP) algorithm by Hernández-Lobato and Adams [30] on Bayesian neural networks. Our experiment settings are almost identity, except that we use a $\text{Gamma}(1, 0.1)$ prior for the inverse covariances and do not use the trick of scaling the input of the output layer. We use neural networks with one hidden layers, and take 50 hidden units for most datasets, except that we take 100 units for Protein and Year which are relatively large; all the datasets are randomly partitioned into $90\%$ for training and $10\%$ for testing, and the results are averaged over 20 random trials, except for Protein and Year on which 5 and 1 trials are repeated, respectively. We use $\text{RELU}(x) = \max(0, x)$ as the active function, whose weak derivative is $\mathbb{I}[x > 0]$ (Stein's identity also holds for weak derivatives; see Stein et al. [31]). PBP is repeated using the default setting of the authors' code[6]. For our algorithm, we only use 20 particles, and use AdaGrad with momentum as what is standard in deep learning. The mini-batch size is 100 except for Year on which we use 1000.

We find our algorithm consistently improves over PBP both in terms of the accuracy and speed; this is encouraging since PBP were specifically designed for Bayesian neural network. We also find that our results are comparable with the more recent results reported on the same datasets [e.g., 32–34] which leverage some advanced techniques that we can also benefit from.

| Dataset | Avg. Test RMSE | | Avg. Test LL | | Avg. Time (Secs) | |
|---|---|---|---|---|---|---|
| | **PBP** | **Our Method** | **PBP** | **Our Method** | **PBP** | **Ours** |
| Boston | $2.977 \pm 0.093$ | $\mathbf{2.957 \pm 0.099}$ | $-2.579 \pm 0.052$ | $\mathbf{-2.504 \pm 0.029}$ | 18 | **16** |
| Concrete | $5.506 \pm 0.103$ | $\mathbf{5.324 \pm 0.104}$ | $-3.137 \pm 0.021$ | $\mathbf{-3.082 \pm 0.018}$ | 33 | **24** |
| Energy | $1.734 \pm 0.051$ | $\mathbf{1.374 \pm 0.045}$ | $-1.981 \pm 0.028$ | $\mathbf{-1.767 \pm 0.024}$ | 25 | **21** |
| Kin8nm | $0.098 \pm 0.001$ | $\mathbf{0.090 \pm 0.001}$ | $0.901 \pm 0.010$ | $\mathbf{0.984 \pm 0.008}$ | 118 | **41** |
| Naval | $0.006 \pm 0.000$ | $\mathbf{0.004 \pm 0.000}$ | $3.735 \pm 0.004$ | $\mathbf{4.089 \pm 0.012}$ | 173 | **49** |
| Combined | $4.052 \pm 0.031$ | $\mathbf{4.033 \pm 0.033}$ | $-2.819 \pm 0.008$ | $\mathbf{-2.815 \pm 0.008}$ | 136 | **51** |
| Protein | $4.623 \pm 0.009$ | $\mathbf{4.606 \pm 0.013}$ | $-2.950 \pm 0.002$ | $\mathbf{-2.947 \pm 0.003}$ | 682 | **68** |
| Wine | $0.614 \pm 0.008$ | $\mathbf{0.609 \pm 0.010}$ | $-0.931 \pm 0.014$ | $\mathbf{-0.925 \pm 0.014}$ | 26 | **22** |
| Yacht | $\mathbf{0.778 \pm 0.042}$ | $0.864 \pm 0.052$ | $\mathbf{-1.211 \pm 0.044}$ | $-1.225 \pm 0.042$ | 25 | 25 |
| Year | $8.733 \pm \text{NA}$ | $\mathbf{8.684 \pm \text{NA}}$ | $-3.586 \pm \text{NA}$ | $\mathbf{-3.580 \pm \text{NA}}$ | 7777 | **684** |

## 6 Conclusion

We propose a simple general purpose variational inference algorithm for fast and scalable Bayesian inference. Future directions include more theoretical understanding on our method, more practical applications in deep learning models, and other potential applications of our basic Theorem in Section 3.1.

**Acknowledgement** This work is supported in part by NSF CRII 1565796.

## Footnotes

[1]code: http://www.cs.princeton.edu/ mdhoffma/

[2]code: http://gershmanlab.webfactional.com/pubs/npv.v1.zip

[3]`https://www.csie.ntu.edu.tw/~cjlin/libsvmtools/datasets/binary.html`

[4]code: `http://www.aueb.gr/users/mtitsias/code/dsvi_matlabv1.zip`.

[5]We scale the gradient of SGLD by a factor of $1/n$ to make it match with the scale of our gradient in (8).

[6]https://github.com/HIPS/Probabilistic-Backpropagation

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
