[Supplementary Material · 1239-appendix.pdf]

# Appendix

**Qiang Liu**      **Dilin Wang**
Department of Computer Science
Dartmouth College
Hanover, NH 03755
{qiang.liu, dilin.wang.gr}@dartmouth.edu

## A  Proof of Theorem 3.1

**Lemma A.1.** *Let $q$ and $p$ be two smooth densities, and $\boldsymbol{T} = \boldsymbol{T}_\epsilon(x)$ an one-to-one transform on $\mathcal{X}$ indexed by parameter $\epsilon$, and $\boldsymbol{T}$ is differentiable w.r.t. both $x$ and $\epsilon$. Define $q_{[\boldsymbol{T}]}$ to be the density of $z = \boldsymbol{T}_\epsilon(x)$ when $x \sim q$, and $\boldsymbol{s}_p(x) = \nabla_x \log p(x)$, we have*

$$\nabla_\epsilon \mathrm{KL}(q_{[\boldsymbol{T}]} \,||\, p) = \mathbb{E}_q \big[ \boldsymbol{s}_p(\boldsymbol{T}(x))^\top \nabla_\epsilon \boldsymbol{T}(x) + \mathrm{trace}((\nabla_x \boldsymbol{T}(x))^{-1} \cdot \nabla_\epsilon \nabla_x \boldsymbol{T}(x)) \big].$$

*Proof.* Denote by $q_{[\boldsymbol{T}^{-1}]}(z)$ the density of $z = \boldsymbol{T}^{-1}(x)$ when $x \sim q(x)$, then

$$q_{[\boldsymbol{T}^{-1}]}(x) = q(\boldsymbol{T}(x)) \cdot |\det(\nabla_x \boldsymbol{T}(x))|.$$

By the change of variable, we have

$$\mathrm{KL}(q_{[\boldsymbol{T}]} \,||\, p) = \mathrm{KL}(q \,||\, p_{[\boldsymbol{T}^{-1}]}),$$

and hence

$$\nabla_\epsilon \mathrm{KL}(q_{[\boldsymbol{T}]} \,||\, p) = -\mathbb{E}_{x \sim q}[\nabla_\epsilon \log p_{[\boldsymbol{T}^{-1}]}(x)].$$

We just need to calculate $\log p_{[\boldsymbol{T}^{-1}]}(x)$; define $\boldsymbol{s}_p(x) = \nabla_x \log p(x)$, we get

$$\nabla_\epsilon \log p_{[\boldsymbol{T}^{-1}]}(x) = \boldsymbol{s}_p(\boldsymbol{T}(x))^\top \nabla_\epsilon \boldsymbol{T}(x) + \mathrm{trace}((\nabla_x \boldsymbol{T}(x))^{-1} \cdot \nabla_\epsilon \nabla_x \boldsymbol{T}(x)).$$

$\square$

*Proof of Theorem 3.1.* When $\boldsymbol{T}(x) = x + \epsilon \boldsymbol{\phi}(x)$ and $\epsilon = 0$, we have

$$\boldsymbol{T}(x) = x, \qquad \nabla_\epsilon \boldsymbol{T}(x) = \boldsymbol{\phi}(x), \qquad \nabla_x \boldsymbol{T}(x) = I, \qquad \nabla_\epsilon \nabla_x \boldsymbol{T}(x) = \nabla_x \boldsymbol{\phi}(x),$$

where $I$ is the identity matrix. Using Lemma A.1 gives the result. $\square$

## B  Proof of Theorem 3.3

Assume $\mathcal{H}$ is a scalar-valued RKHS with positive definite kernel $k(x, x')$, then $\mathcal{H}^d = \mathcal{H} \times \cdots \times \mathcal{H}$ is a vector-valued RKHS, which corresponds to a matrix-valued positive definite kernel $\boldsymbol{K}(x, x') = k(x, x')I$, where $I$ is the identity matrix. The reproducing property for this vector-valued RKHS is

$$\boldsymbol{c}^\top \boldsymbol{f}(x) = \langle \boldsymbol{f}(\cdot), \ \boldsymbol{c} k(x, \cdot) \rangle_{\mathcal{H}^d},$$

for $\forall \boldsymbol{f} \in \mathcal{H}^d$ and $\boldsymbol{c} \in \mathbb{R}^d$. Taking derivative on both size, gives

$$\mathrm{trace}(\boldsymbol{C} \nabla_x \boldsymbol{f}(x)) = \langle \boldsymbol{f}(\cdot), \ \boldsymbol{C} \nabla_x k(x, \cdot) \rangle_{\mathcal{H}^d}$$

where $\nabla_x \boldsymbol{f}(x) = [\nabla_x f_1(x), \ldots, \nabla_x f_d(x)]$.

Let $F[\boldsymbol{f}]$ be a functional on $\boldsymbol{f} \in \mathcal{H}^d$. The gradient $\nabla_{\boldsymbol{f}} F[\boldsymbol{f}]$ of $F[\cdot]$ is a function in $\mathcal{H}^d$ that satisfies

$$F[\boldsymbol{f} + \epsilon \boldsymbol{g}] = F[\boldsymbol{f}] + \epsilon \, \langle \nabla_{\boldsymbol{f}} F[\boldsymbol{f}], \, g \rangle_{\mathcal{H}^d} + O(\epsilon^2).$$

*Proof.* Define $F[\boldsymbol{f}] = \mathrm{KL}(q_{[x+\boldsymbol{f}(x)]} \mid\mid p) = \mathrm{KL}(q \mid\mid p_{[(x+\boldsymbol{f}(x))^{-1}]})$, we have

$$
\begin{aligned}
F[f + \epsilon g] &= \mathrm{KL}(q \mid\mid p_{[(x+\boldsymbol{f}(x)+\epsilon\boldsymbol{g}(x))^{-1}]}) \\
&= \mathbb{E}_q[\log q(x) - \log p(x + \boldsymbol{f}(x) + \epsilon\boldsymbol{g}(x)) - \log\det(I + \nabla_x\boldsymbol{f}(x) + \epsilon\nabla_x\boldsymbol{g}(x))],
\end{aligned}
$$

and hence we have

$$
F[\boldsymbol{f} + \epsilon\boldsymbol{g}] - F[\boldsymbol{f}] = -\Delta_1 - \Delta_2,
$$

where

$$
\begin{aligned}
\Delta_1 &= \mathbb{E}_q[\log p(x + \boldsymbol{f}(x) + \epsilon\boldsymbol{g}(x))] - \mathbb{E}_q[\log p(x + \boldsymbol{f}(x))], \\
\Delta_2 &= \mathbb{E}_q[\log\det(I + \nabla_x\boldsymbol{f}(x) + \epsilon\nabla_x\boldsymbol{g}(x))] - \mathbb{E}_q[\log\det(I + \nabla_x\boldsymbol{f}(x))].
\end{aligned}
$$

For the terms in the above equation, we have

$$
\begin{aligned}
\Delta_1 &= \mathbb{E}_q[\log p(x + \boldsymbol{f}(x) + \epsilon\boldsymbol{g}(x))] - \mathbb{E}_q[\log p(x + \boldsymbol{f}(x))] \\
&= \epsilon\,\mathbb{E}_q[\nabla_x\log p(x + \boldsymbol{f}(x))^\top \boldsymbol{g}(x)] + O(\epsilon^2) \\
&= \epsilon\,\mathbb{E}_q[\langle\boldsymbol{g},\ \nabla_x\log p(x + \boldsymbol{f}(x))k(x,\cdot)\rangle_{\mathcal{H}^d}] + O(\epsilon^2) \\
&= \epsilon\,\langle\boldsymbol{g},\ \mathbb{E}_q[\nabla_x\log p(x + \boldsymbol{f}(x))k(x,\cdot)]\rangle_{\mathcal{H}^d} + O(\epsilon^2),
\end{aligned}
$$

and

$$
\begin{aligned}
\Delta_2 &= \mathbb{E}_q[\log\det(I + \nabla_x\boldsymbol{f}(x) + \epsilon\nabla_x\boldsymbol{g}(x))] - \mathbb{E}_q[\log\det(I + \nabla_x\boldsymbol{f}(x))] \\
&= \epsilon\,\mathbb{E}_q[\mathrm{trace}((I + \nabla_x\boldsymbol{f}(x))^{-1}\nabla_x\boldsymbol{g}(x))] + O(\epsilon^2) \\
&= \epsilon\,\mathbb{E}_q[\langle\boldsymbol{g},\ (I + \nabla_x\boldsymbol{f}(x))^{-1}\nabla_x k(x,\cdot)\rangle_{\mathcal{H}^d}] + O(\epsilon^2) \\
&= \epsilon\,\langle\boldsymbol{g},\ \mathbb{E}_q[(I + \nabla_x\boldsymbol{f}(x))^{-1}\nabla_x k(x,\cdot)]\rangle_{\mathcal{H}^d} + O(\epsilon^2)
\end{aligned}
$$

and hence

$$
F[\boldsymbol{f} + \epsilon\boldsymbol{g}] - F[\boldsymbol{f}] = \epsilon\,\langle\nabla_{\boldsymbol{f}}F[\boldsymbol{f}],\ \boldsymbol{g}\rangle_{\mathcal{H}^d} + O(\epsilon^2),
$$

where

$$
\nabla_{\boldsymbol{f}}F[\boldsymbol{f}] = -\mathbb{E}_q[\nabla_x\log p(x + \boldsymbol{f}(x))k(x,\cdot) + (I + \nabla_x\boldsymbol{f}(x))^{-1}\nabla_x k(x,\cdot)]. \tag{B.1}
$$

Taking $\boldsymbol{f} = 0$ then gives the desirable result. $\qquad\square$

## C   Connection with de Bruijn's identity and Fisher Divergence

If we take $\phi_{q,p}(x) = \nabla_x\log p(x) - \nabla_x\log q(x)$ in (5), we can show that (5) reduces to

$$
\nabla_\epsilon\mathrm{KL}(q_{[\boldsymbol{T}]} \mid\mid p)\big|_{\epsilon=0} = -\mathbb{F}(q,\ p),
$$

where $\mathbb{F}(q,\ p)$ is the Fisher divergence between $p$ and $q$, defined as

$$
\mathbb{F}(q,\ p) = \mathbb{E}_q[||\nabla_x\log p - \nabla_x\log q||_2^2].
$$

Note that this can be treated as a deterministic version of *de Bruijn's identity* (Cover and Thomas, 2012; Lyu, 2009), which draws similar connection between KL and Fisher divergence, but uses randomized linear transform $\boldsymbol{T}(x) = x + \sqrt{\epsilon}\cdot\xi$, where $\xi$ is a standard Gaussian noise. Close connections can also be drawn with Langevin dynamics, which we will elaborate in future works.

## D   Additional Experiments

We collect additional experimental results that can not fitted into the main paper.

### D.1 Bayesian Logistic Regression on Small Datasets

We consider the Bayesian logistic regression model for binary classification, on which the regression weights $w$ is assigned with a Gaussian prior $p_0(w) = \mathcal{N}(w, \alpha^{-1})$ and $p_0(\alpha) = \Gamma(\alpha, a, b)$, and apply inference on posterior $p(x \mid D)$, where $x = [w, \log \alpha]$. The hyper-parameter is taken to be $a = 1$ and $b = 0.01$. This setting is the same as that in Gershman et al. (2012). We compared our algorithm with the no-U-turn sampler (NUTS)[1] (Hoffman and Gelman, 2014) and non-parametric variational inference (NPV)[2] on the 8 datasets ($N > 500$) as used in Gershman et al. (2012), in which we use 100 particles, NPV uses 100 mixture components, and NUTS uses 1000 draws with 1000 burnin period. We find that all these three algorithms almost always performs the same across the 8 datasets (See Figure   in Appendix), and this is consistent with Figure 2 of Gershman et al. (2012).

We further experimented on a toy dataset with only two features and visualize the prediction probability of the three algorithms in Figure D.1. We again find that all the three algorithms tend to perform similarly. Note, however, that NPV is relatively inconvenient to use since it requires the Hessian matrix, and NUTS tends to be very small when applied on massive datasets.

(a) Testing Accuracy  (b) Testing Log-Likelihood

Figure 1: Bayesian logistic regression on the 8 datasets studied in Gershman et al. (2012). We find our method performs similarly as NPV and NUTS on all the 8 datasets.

Figure 2: Bayesian logistic regression. The posterior prediction uncertainty as inferred by different approaches on a toy data.

## Footnotes

[1]code: http://www.cs.princeton.edu/ mdhoffma/

[2]code: http://gershmanlab.webfactional.com/pubs/npv.v1.zip