[Reviews · NeurIPS 2016]

Reviewer 1

Summary

This paper proposes a new technique for general variational inference. In contrast to classical variational inference, the authors consider adapting the shape and properties of the variational approximation while minimizing its KL divergence to the true posterior. The technique relies on iteratively transforming particles, motivated by a connection between the KL divergence and a kernelized Stein discrepancy. The paper presents both theoretical and experimental results.

Qualitative Assessment

General comments: This is a lovely paper. Thank you for submitting it. Overall I'm quite satisfied with what the paper presents and how it is laid out. I think the paper offers a good balance of new ideas, theory, intuition, and experimental results. My only concern is sensitivity to bandwidth `h` and number of particles `n`. I like how the authors present an adaptive scheme for `h`, yet was a bit concerned about the `0.002` factor necessary for the Bayesian Logistic Regression example. It would be valuable to develop a similar rule of thumb for `n` too. Minor comments: + line 42: broken sentence + line 46: this citation is from the future? :) + line 72: perhaps consider a simple example here, for the uninitiated reader? + line 75: i must admit, i had a bit of trouble following this narrative. the subtlety between a non-kernelized and a kernelized Stein discrepancy is still a bit unclear to me. + line 85: should this be S(q, p) instead of the other way around? + eq 5: why no square brackets for the entropy of q? + line 113: broken sentence + line 116: broken sentence + line 162: i thought that this remark could have gone into the appendix. then you could afford a longer conclusion. + line 187: what is the "full Bayesian approach"? i understand what you're saying here, but perhaps consider a more explicit statement? + line 265: Hoffman and Gelman typo + line 286: Adagrad typo

Confidence in this Review

2-Confident (read it all; understood it all reasonably well)


Reviewer 2

Summary

This paper describes a general method that can be used for approximate inference. The technique consists in iteratively applying a non-linear transformation to several samples from a starting distribution q0, to enforce that the distribution of the corresponding samples is closer to that of a target distribution p. After enough of these transformation have been made, the samples can be used to compute expectations with respect to the target distribution. The proposed method is illustrated in a logistic regression model and in Bayesian neural networks.

Qualitative Assessment

I really like the idea of using the Stein discrepancy for doing approximate inference. I also like the idea of having a set of particles (samples) that are iteratively transformed to match the target distribution. However, I believe that the writing of the paper has to be improved. At this point it is very difficult to follow Section 2, which has been written in a sloppy way. In particular, I have found much more rigorous the reference by Liu et al. Is Eq. (3) missing an expectation? How do you obtain (4)? Has that been shown elsewhere, if so a reference should be given. The format employed for the references is incorrect. It should use numbers instead of author's name + year. The experimental section should compare the proposed approach with other related methods described in Section 4. In particular, the other methods related to variable transformations described there.

Confidence in this Review

2-Confident (read it all; understood it all reasonably well)


Reviewer 3

Summary

The authors propose a variational inference algorithm for smooth densities based on functional gradient descent of the KLD(q,p) in RKHS. On practice the posterior density is approximated by a finite set of particles and updates are performed directly to the particles. Performance is evaluated on one toy unsupervised domain and two larger supervised domains. Comparisons with relevant methods are provided.

Qualitative Assessment

The proposed algorithm extends and generalizes the idea of normalizing flows to a non-parametric setting. The connection of functional gradients of the KLD with respect to the transformations in RKHS and Stein discrepancy is very interesting and provides a bridge between variational and kernel methods. Main points: Overall the paper is well structured, the derivations are sound (with a few typos). The typos in the formulas made it hard to read at first sight. To the best of my knowledge* the main contribution of this paper was to provide a new bridge between kernel and variational methods. It is somewhat unsatisfying that the final algorithm turns out to be based on particle-optimization instead of representing smooth functions directly. On another hand, the proposed algorithm is surprisingly simple to implement. The authors provided classification accuracy comparisons to other methods (and predictive uncertainty in the supplements), but only a toy example was provided to assess the quality of the posterior approximation. It would strengthen the paper if the authors have provided more concrete analysis of the quality of the posterior approximations learned by the model in more interesting domains. Minor points: (1) The transpose in equation (2) should be on \phi(x) instead of \nabla_x log p(x) (2) There is an expectation under q(x) missing in equation (3) (3) Line 142 (last equation). Shouldn't there be a square root in S?: \nabla_{\epsilon}KL = -S^{1/2} (4) In algorithm (1) there should be no index i in \phi(x_i) on the l.h.s of (9). (5) Line 230: " … where h is the ..." should be "... where med is the ..." * I am not entirely familiar with the literature on kernel methods

Confidence in this Review

2-Confident (read it all; understood it all reasonably well)


Reviewer 4

Summary

The authors derive a new inference algorithm for bayesian inference connecting variational inference to recent results using Kernelized Stein discrepancy. The algorithm is particle based and smoothly interpolates between MAP and true Bayesian posterior (by using more particles). At a high level, the authors connect the derivative of a small perturbation (x+eps phi(x), which are going to be progressively used to transform an initial distribution into the target distribution) to the stein operator. Then, by restricting the deformation from identity to be a function in a RKHS, they can obtain a closed-form for the optimal deformation function phi, which takes the form of a stein operator on the initial distribution. Distributions are represented by samples and updated according to the simple operator.

Qualitative Assessment

Overall, I found the paper interesting; the paper offers new theory as well as numerical results comparable to the state of the art on decently difficult datasets. Perhaps due to space constraints, an important part of the paper (section 3.2) - the inference algorithm - is poorly explained. In particular, I initially thought that the use of particles meant that the approximating distribution was a sum of Dirac delta functions - but that cannot be the case since, even with many particles, the 'posterior' would degenerate into the MAP (note that in similar work, authors either use particles when p(x) involves discrete x variables, as in Kulkarni et al, or 'smooth' the particles to approximate a continuous distribution, as in Gershman et al). Instead, it looks like the algorithm works directly on samples of the distribution q0, q1.. (hence the vague 'for whatever distribution q that {xi}ni=1 currently represents'). It is tempting to consider q_i to be a kernel density estimate (mixture of normals with fixed width), and see if we can approximate equation 9 for that representation to be stable. Is this possible / something the authors have considered? Also, since the algorithm does not care about what precise form the distribution q takes, but the algorithm cannot possibly work for q sum of delta functions, I assume there are some smoothness assumptions required for the theorem to hold (even if it is simple as 'q has a density'). Minor: - In the second paragraph of the introduction, the authors explain that in some work (Kingma and Welling, Rezende and Mohamed, etc.), neural networks are used to approximate a posterior distribution; this is not accurate: in those papers the posterior is typically a simple distribution (often, gaussian mean-field); the neural network is used to amortize inference and predict the parameters of the posterior as a function of the data. This is faster, but theoretically worse than regular mean-field inference (optimization issues put aside). Rezende and Mohamed 2015 is, however, truly using a neural network to parametrize a more complex posterior distribution. - The connection to the reparametrization trick is very promising and would have deserved a few words (even if future work goes into mode details), at least in the appendix, where the reparametrization is key in obtaining 3.1.

Confidence in this Review

2-Confident (read it all; understood it all reasonably well)


Reviewer 5

Summary

The authors propose a form of functional gradient descent for variational inference, leveraging ideas from the kernelized Stein discrepancy to improve upon the approximation quality in variational inference methods.

Qualitative Assessment

The paper makes very interesting connections between kernelized Stein discrepancies and KL minimization. It derives useful properties out of these connections to be used as part of a variational inference algorithm in practice. However, the experiments are limited, using toy models such as Bayesian logistic regression, with the most complex model being a Bayesian neural network applied to toy data sets from UCI. It is also lacking in comparison to other variational inference methods. The baseline in the Bayesian neural network example uses a very simple approximating distribution, which in recent years we have improved, e.g., Rezende et al. (2015); Louizos et al. (2016). In comparison to other developments in expressive variational families, the efficiency of the algorithm is strongly hindered by the number of particles, which can in practice make it either slower or a poorer approximation than non-mean-field approaches. This makes empirical comparisons crucial for understanding how useful the approach may be in practice. Other comments + Variational gradient descent is a broad and confusing term, as it can be confused with the standard paradigm of variational inference using gradient descent, or gradient descent as some form of variational inference (e.g., Mandt et al. (2016)). This is perhaps worth a rename. + The paper argues that the algorithm makes no parametric assumptions, although the variational family used for the approximation is implicit under the finite set of particles, just in the same way that the other particle approximation approaches for variational inference interpret the particles as forming a (possibly weighted) mixture of point mass distributions. + lines 24-26 and 102-103: The variational distributions built with neural networks in the referenced works do not correspond to more complex approximations (only Rezende and Mohamed (2015) does). Many in fact simplify the approximation in order to amortize inference, actually making optimization easier in practice. The works referenced for more complex distributions should refer to other work, e.g., by Dinh et al. (2014); Salimans et al. (2015); Tran et al. (2016); Ranganath et al. (2016). + lines 112-116: The core idea here has been used in the context of recent variational inference works by Tran et al. (2016). This sounds like a useful reference and source of comparison.

Confidence in this Review

3-Expert (read the paper in detail, know the area, quite certain of my opinion)


Reviewer 6

Summary

The authors present a new particle variational approximation algorithm based on a connection between the kernel stein operator and the KL divergence. With this connection recent results on computing the kernel stein discrepancy can be used to find a direction (a function the Hilbert space) which minimizes the KL divergence locally. The authors evaluate their algorithm on logistic regression and a small neural network.

Qualitative Assessment

The connection the author’s present allows the to leverage the recent results on kernel stein discrepancies to develop a simple algorithm for particle variational inference that does not require further approximation. My main concern lies the in applicability of this work. The authors have not explored problems of the size that truly necessitate variational inference. It’s important to me to see results on larger models like multilayer DGLMS or DRAW for images or maybe deep hierarchies of gamma random variables where sampling procedures are impractical. I’ve listed some more detailed comments below: 1) Do you need finiteness for the definitions in 55-59? 2) The requirements on the best set of Q seem unachievable for general models. Universality and tractability oppose each other. That is having a universal family with tractable updates does not mean it’s tractable to find the posterior in Q. How can you better specify the requirements to resolve this tension? 3) The comments around 119-122 seem to miss out on recent work that introduces latent variables into the variation approximation [see “ Markov Chain Monte Carlo and Variational Inference: Bridging the Gap” at (ICML 2015) or “Auxiliary Deep Generative Models” or “Hierarchical Variational Models” both at (ICML 2016)]. Discussion of this line of work is missing from the paper 4) Explain further the use of the implicit function theorem at line 132 5) The comments around the n^2 complexity and the relative costs in 194-197 need to be explored further. In practice, the number of particles needed likely scales with the model complexity which also scales the gradient compute time. 6) The selection of the bandwidth parameter feels ad hoc in the experiments. This parameters seems very important. Can this be optimized as a variational parameter? It feels like that leads to a form of overfitting 7) There are a bunch of typos. See for example line 338, “Homan” should be “Hoffman” 8) The discussion of the use of neural networks feels hand wavy in the introduction.

Confidence in this Review

2-Confident (read it all; understood it all reasonably well)